# Stabilizing Training of Generative Adversarial Networks through Regularization

**Kevin Roth**
Department of Computer Science
ETH Zürich
kevin.roth@inf.ethz.ch

**Aurelien Lucchi**
Department of Computer Science
ETH Zürich
aurelien.lucchi@inf.ethz.ch

**Sebastian Nowozin**
Microsoft Research
Cambridge, UK
sebastian.Nowozin@microsoft.com

**Thomas Hofmann**
Department of Computer Science
ETH Zürich
thomas.hofmann@inf.ethz.ch

## Abstract

Deep generative models based on Generative Adversarial Networks (GANs) have demonstrated impressive sample quality but in order to work they require a careful choice of architecture, parameter initialization, and selection of hyper-parameters. This fragility is in part due to a dimensional mismatch or non-overlapping support between the model distribution and the data distribution, causing their density ratio and the associated $f$-divergence to be undefined. We overcome this fundamental limitation and propose a new regularization approach with low computational cost that yields a stable GAN training procedure. We demonstrate the effectiveness of this regularizer across several architectures trained on common benchmark image generation tasks. Our regularization turns GAN models into reliable building blocks for deep learning. [1]

## 1 Introduction

A recent trend in the world of generative models is the use of deep neural networks as data generating mechanisms. Two notable approaches in this area are variational auto-encoders (VAEs) [14, 28] as well as generative adversarial networks (GAN) [8]. GANs are especially appealing as they move away from the common likelihood maximization viewpoint and instead use an adversarial game approach for training generative models. Let us denote by $\mathbb{P}(\mathbf{x})$ and $\mathbb{Q}_\theta(\mathbf{x})$ the data and model distribution, respectively. The basic idea behind GANs is to pair up a $\theta$-parametrized generator network that produces $\mathbb{Q}_\theta$ with a discriminator which aims to distinguish between $\mathbb{P}$ and $\mathbb{Q}_\theta$, whereas the generator aims for making $\mathbb{Q}_\theta$ indistinguishable from $\mathbb{P}$. Effectively the discriminator represents a class of objective functions $\mathcal{F}$ that measures dissimilarity of pairs of probability distributions. The final objective is then formed via a supremum over $\mathcal{F}$, leading to the saddle point problem

$$\min_\theta \left[ \ell(\mathbb{Q}_\theta; \mathcal{F}) := \sup_{F \in \mathcal{F}} F\left(\mathbb{P}, \mathbb{Q}_\theta\right) \right]. \tag{1}$$

The standard way of representing a specific $\mathcal{F}$ is through a family of statistics or discriminants $\phi \in \Phi$, typically realized by a neural network [8, 26]. In GANs, we use these discriminators in a logistic classification loss as follows

$$F(\mathbb{P}, \mathbb{Q}; \phi) = \mathbf{E}_\mathbb{P}\left[g(\phi(\mathbf{x}))\right] + \mathbf{E}_\mathbb{Q}\left[g(-\phi(\mathbf{x}))\right], \tag{2}$$

where $g(z) = \ln(\sigma(z))$ is the log-logistic function (for reference, $\sigma(\phi(\mathbf{x})) = D(\mathbf{x})$ in [8]).

As shown in [8], for the Bayes-optimal discriminator $\phi^* \in \Phi$, the above generator objective reduces to the Jensen-Shannon (JS) divergence between $\mathbb{P}$ and $\mathbb{Q}$. The work of [25] later generalized this to a more general class of $f$-divergences, which gives more flexibility in cases where the generative model may not be expressive enough or where data may be scarce.

We consider three different challenges for learning the model distribution:

(A) *empirical estimation:* the model family may contain the true distribution or a good approximation thereof, but one has to identify it based on a finite training sample drawn from $\mathbb{P}$. This is commonly addressed by the use of regularization techniques to avoid overfitting, e.g. in the context of estimating $f$-divergences with $M$-estimators [24]. In our work, we suggest a novel (Tikhonov) regularizer, derived and motivated from a training-with-noise scenario, where $\mathbb{P}$ and $\mathbb{Q}$ are convolved with white Gaussian noise [30, 3], namely

$$F_\gamma(\mathbb{P}, \mathbb{Q}; \phi) := F(\mathbb{P} * \Lambda, \mathbb{Q} * \Lambda; \phi), \quad \Lambda = \mathcal{N}(\mathbf{0}, \gamma \mathbf{I}). \tag{3}$$

(B) *density misspecification:* the model distribution and true distribution both have a density function with respect to the same base measure but there exists no parameter for which these densities are sufficiently similar. Here, the principle of parameter estimation via divergence minimization is provably sound in that it achieves a well-defined limit [1, 21]. It therefore provides a solid foundation for statistical inference that is robust with regard to model misspecifications.

(C) *dimensional misspecification:* the model distribution and the true distribution do not have a density function with respect to the same base measure or – even worse – $\mathrm{supp}(\mathbb{P}) \cap \mathrm{supp}(\mathbb{Q})$ may be negligible. This may occur, whenever the model and/or data are confined to low-dimensional manifolds [3, 23]. As pointed out in [3], a geometric mismatch can be detrimental for $f$-GAN models as the resulting $f$-divergence is not finite (the sup in Eq. (1) is $+\infty$). As a remedy, it has been suggested to use an alternative family of distance functions known as *integral probability metrics* [22, 31]. These include the Wasserstein distance used in Wasserstein GANs (WGAN) [3] as well as RKHS-induced maximum mean discrepancies [9, 16, 6], which all remain well-defined. We will provide evidence (analytically and experimentally) that the noise-induced regularization method proposed in this paper effectively makes $f$-GAN models robust against dimensional misspecifications. While this introduces some dependency on the (Euclidean) metric of the ambient data space, it does so on a well-controlled length scale (the amplitude of noise or strength of the regularization $\gamma$) and by retaining the benefits of $f$-divergences. This is a rather gentle modification compared to the more radical departure taken in Wasserstein GANs, which rely solely on the ambient space metric (through the notion of optimal mass transport).

In what follows, we will take Eq. (3) as the starting point and derive an approximation via a regularizer that is simple to implement as an integral operator penalizing the squared gradient norm. As opposed to a naïve norm penalization, each $f$-divergence has its own characteristic weighting function over the input space, which depends on the discriminator output. We demonstrate the effectiveness of our approach on a simple Gaussian mixture as well as on several benchmark image datasets commonly used for generative models. In both cases, our proposed regularization yields stable GAN training and produces samples of higher visual quality. We also perform pairwise tests of regularized vs. unregularized GANs using a novel *cross-testing protocol*.

In summary, we make the following contributions:

- We systematically derive a novel, efficiently computable regularization method for $f$-GAN.
- We show how this addresses the dimensional misspecification challenge.
- We empirically demonstrate stable GAN training across a broad set of models.

## 2 Background

The fundamental way to learn a generative model in machine learning is to (i) define a parametric family of probability densities $\{\mathbb{Q}_\theta\}$, $\theta \in \Theta \subseteq \mathbb{R}^d$, and (ii) find parameters $\theta^* \in \Theta$ such that $\mathbb{Q}_\theta$ is closest (in some sense) to the true distribution $\mathbb{P}$. There are various ways to measure how close model and real distribution are, or equivalently, various ways to define a distance or divergence function between $\mathbb{P}$ and $\mathbb{Q}$. In the following we review different notions of divergences used in the literature.

$f$-**divergence.** GANs [8] are known to minimize the Jensen-Shannon divergence between $\mathbb{P}$ and $\mathbb{Q}$. This was generalized in [25] to $f$-divergences induced by a convex functions $f$. An interesting property of $f$-divergences is that they permit a variational characterization [24, 27] via

$$D_f(\mathbb{P}||\mathbb{Q}) := \mathbf{E}_{\mathbb{Q}}\left[f \circ \frac{d\mathbb{P}}{d\mathbb{Q}}\right] = \int_{\mathcal{X}} \sup_u \left(u \cdot \frac{d\mathbb{P}}{d\mathbb{Q}} - f^c(u)\right) d\mathbb{Q}, \tag{4}$$

where $d\mathbb{P}/d\mathbb{Q}$ is the Radon-Nikodym derivative and $f^c(t) \equiv \sup_{u \in \text{dom}_f}\{ut - f(u)\}$ is the *Fenchel dual* of $f$. By defining an arbitrary class of statistics $\Psi \ni \psi : \mathcal{X} \to \mathbb{R}$ we arrive at the bound

$$D_f(\mathbb{P}||\mathbb{Q}) \geq \sup_\psi \int \left(\psi \cdot \frac{d\mathbb{P}}{d\mathbb{Q}} - f^c \circ \psi\right) d\mathbb{Q} = \sup_\psi \left\{\mathbf{E}_{\mathbb{P}}[\psi] - \mathbf{E}_{\mathbb{Q}}[f^c \circ \psi]\right\}. \tag{5}$$

Eq. (5) thus gives us a variational lower bound on the $f$-divergence as an expectation over $\mathbb{P}$ and $\mathbb{Q}$, which is easier to evaluate (e.g. via sampling from $\mathbb{P}$ and $\mathbb{Q}$, respectively) than the density based formulation. We can see that by identifying $\psi = g \circ \phi$ and with the choice of $f$ such that $f^c = -\ln(1 - \exp)$, we get $f^c \circ \psi = -\ln(1 - \sigma(\phi)) = -g(-\phi)$ thus recovering Eq. (2).

**Integral Probability Metrics (IPM).** An alternative family of divergences are integral probability metrics [22, 31], which find a witness function to distinguish between $\mathbb{P}$ and $\mathbb{Q}$. This class of methods yields an objective similar to Eq. (2) that requires optimizing a distance function between two distributions over a function class $\mathcal{F}$. Particular choices for $\mathcal{F}$ yield the kernel maximum mean discrepancy approach of [9, 16] or Wasserstein GANs [3]. The latter distance is defined as

$$W(\mathbb{P}, \mathbb{Q}) = \sup_{\|f\|_L \leq 1} \left\{\mathbf{E}_{\mathbb{P}}[f] - \mathbf{E}_{\mathbb{Q}}[f]\right\}, \tag{6}$$

where the supremum is taken over functions $f$ which have a bounded Lipschitz constant.

As shown in [3], the Wasserstein metric implies a different notion of convergence compared to the JS divergence used in the original GAN. Essentially, the Wasserstein metric is said to be *weak* as it requires the use of a weaker topology, thus making it easier for a sequence of distribution to converge. The use of a weaker topology is achieved by restricting the function class to the set of bounded Lipschitz functions. This yields a hard constraint on the function class that is empirically hard to satisfy. In [3], this constraint is implemented via weight clipping, which is acknowledged to be a "terrible way" to enforce the Lipschitz constraint. As will be shown later, our regularization penalty can be seen as a soft constraint on the Lipschitz constant of the function class which is easy to implement in practice. Recently, [10] has also proposed a similar regularization; while their proposal was motivated for Wasserstein GANs and does not extend to $f$-divergences it is interesting to observe that both their and our regularization work on the gradient.

**Training with Noise.** As suggested in [3, 30], one can break the dimensional misspecification discussed in Section 1 by adding continuous noise to the inputs of the discriminator, therefore smoothing the probability distribution. However, this requires to add high-dimensional noise, which introduces significant variance in the parameter estimation process. Counteracting this requires a lot of samples and therefore ultimately leads to a costly or impractical solution. Instead we propose an approach that relies on analytic convolution of the densities $\mathbb{P}$ and $\mathbb{Q}$ with Gaussian noise. As we demonstrate below, this yields a simple weighted penalty function on the norm of the gradients. Conceptually we think of this noise not as being part of the generative process (as in [3]), but rather as a way to define a smoother family of discriminants for the variational bound of $f$-divergences.

**Regularization for Mode Dropping.** Other regularization techniques address the problem of mode dropping and are complementary to our approach. This includes the work of [7] which incorporates a supervised training signal as a regularizer on top of the discriminator target. To implement supervision the authors use an additional auto-encoder as well as a two-step training procedure which might be computationally expensive. A similar approach was proposed by [20] that stabilizes GANs by unrolling the optimization of the discriminator. The main drawback of this approach is that the computational cost scales with the number of unrolling steps. In general, it is not clear to what extent these methods not only stabilize GAN training, but also address the conceptual challenges listed in Section 1.

## 3 Noise-Induced Regularization

From now onwards, we consider the general $f$-GAN [25] objective defined as

$$F(\mathbb{P}, \mathbb{Q}; \psi) \equiv \mathbf{E}_{\mathbb{P}}[\psi] - \mathbf{E}_{\mathbb{Q}}[f^c \circ \psi]. \tag{7}$$

### 3.1 Noise Convolution

From a practitioners point of view, training with noise can be realized by adding zero-mean random variables $\boldsymbol{\xi}$ to samples $\mathbf{x} \sim \mathbb{P}, \mathbb{Q}$ during training. Here we focus on normal white noise $\boldsymbol{\xi} \sim \Lambda = \mathcal{N}(\mathbf{0}, \gamma \mathbf{I})$ (the same analysis goes through with a Laplacian noise distribution for instance). From a theoretical perspective, adding noise is tantamount to convolving the corresponding distribution as

$$\mathbf{E}_{\mathbb{P}} \mathbf{E}_{\Lambda}[\psi(\mathbf{x} + \boldsymbol{\xi})] = \int \psi(\mathbf{x}) \int p(\mathbf{x} - \boldsymbol{\xi}) \lambda(\boldsymbol{\xi}) d\boldsymbol{\xi} \, d\mathbf{x} = \int \psi(\mathbf{x})(p * \lambda)(\mathbf{x}) d\mathbf{x} = \mathbf{E}_{\mathbb{P}*\Lambda}[\psi]. \tag{8}$$

where $p$ and $\lambda$ are probability densities of $\mathbb{P}$ and $\Lambda$, respectively, with regard to the Lebesgue measure. The noise distribution $\Lambda$ as well as the resulting $\mathbb{P}*\Lambda$ are guaranteed to have full support in the ambient space, i.e. $\lambda(\mathbf{x}) > 0$ and $(p * \lambda)(\mathbf{x}) > 0$ $(\forall \mathbf{x})$. Technically, applying this to both $\mathbb{P}$ and $\mathbb{Q}$ makes the resulting generalized $f$-divergence well-defined, even when the generative model is dimensionally misspecified. Note that approximating $\mathbf{E}_{\Lambda}$ through sampling was previously investigated in [30, 3].

### 3.2 Convolved Discriminants

With symmetric noise, $\lambda(\boldsymbol{\xi}) = \lambda(-\boldsymbol{\xi})$, we can write Eq. (8) equivalently as

$$\mathbf{E}_{\mathbb{P}*\Lambda}[\psi] = \mathbf{E}_{\mathbb{P}} \mathbf{E}_{\Lambda}[\psi(\mathbf{x} + \boldsymbol{\xi})] = \int p(\mathbf{x}) \int \psi(\mathbf{x} - \boldsymbol{\xi}) \lambda(-\boldsymbol{\xi}) \, d\boldsymbol{\xi} \, d\mathbf{x} = \mathbf{E}_{\mathbb{P}}[\psi * \lambda]. \tag{9}$$

For the $\mathbb{Q}$-expectation in Eq. (7) one gets, by the same argument, $\mathbf{E}_{\mathbb{Q}*\Lambda}[f^c \circ \psi] = \mathbf{E}_{\mathbb{Q}}[(f^c \circ \psi) * \lambda]$. Formally, this generalizes the variational bound for $f$-divergences in the following manner:

$$F(\mathbb{P} * \Lambda, \mathbb{Q} * \Lambda; \psi) = F(\mathbb{P}, \mathbb{Q}; \psi * \lambda, (f^c \circ \psi) * \lambda), \quad F(\mathbb{P}, \mathbb{Q}; \rho, \tau) := \mathbf{E}_{\mathbb{P}}[\rho] - \mathbf{E}_{\mathbb{Q}}[\tau] \tag{10}$$

Assuming that $\mathcal{F}$ is closed under $\Lambda$ convolutions, the regularization will result in a relative weakening of the discriminator as we take the sup over a smaller, more regular family. Clearly, the low-pass effect of $\Lambda$-convolutions can be well understood in the Fourier domain. In this equivalent formulation, we leave $\mathbb{P}$ and $\mathbb{Q}$ unchanged, yet we change the view the discriminator can take on the ambient data space: metaphorically speaking, the generator is paired up with a short-sighted adversary.

### 3.3 Analytic Approximations

In general, it may be difficult to analytically compute $\psi * \lambda$ or – equivalently – $\mathbf{E}_{\Lambda}[\psi(\mathbf{x} + \boldsymbol{\xi})]$. However, for small $\gamma$ we can use a Taylor approximation of $\psi$ around $\boldsymbol{\xi} = 0$ (cf. [5]):

$$\psi(\mathbf{x} + \boldsymbol{\xi}) = \psi(\mathbf{x}) + [\nabla \psi(\mathbf{x})]^T \boldsymbol{\xi} + \frac{1}{2} \boldsymbol{\xi}^T [\nabla^2 \psi(\mathbf{x})] \boldsymbol{\xi} + \mathcal{O}(\boldsymbol{\xi}^3) \tag{11}$$

where $\nabla^2$ denotes the Hessian, whose trace $\text{Tr}(\nabla^2) = \triangle$ is known as the Laplace operator. The properties of white noise result in the approximation

$$\mathbf{E}_{\Lambda}[\psi(\mathbf{x} + \boldsymbol{\xi})] = \psi(\mathbf{x}) + \frac{\gamma}{2} \triangle \psi(\mathbf{x}) + \mathcal{O}(\gamma^2) \tag{12}$$

and thereby lead directly to an approximation of $F_{\gamma}$ (see Eq. (3)) via $F = F_0$ plus a correction, i.e.

$$F_{\gamma}(\mathbb{P}, \mathbb{Q}; \psi) = F(\mathbb{P}, \mathbb{Q}; \psi) + \frac{\gamma}{2} \{ \mathbf{E}_{\mathbb{P}}[\triangle \psi] - \mathbf{E}_{\mathbb{Q}}[\triangle(f^c \circ \psi)] \} + \mathcal{O}(\gamma^2). \tag{13}$$

We can interpret Eq. (13) as follows: the Laplacian measures how much the scalar fields $\psi$ and $f^c \circ \psi$ differ at each point from their local average. It is thereby an infinitesimal proxy for the (exact) convolution.

The Laplace operator is a sum of $d$ terms, where $d$ is the dimensionality of the ambient data space. As such it does not suffer from the quadratic blow-up involved in computing the Hessian. If we realize the discriminator $\psi$ via a deep network, however, then we need to be able to compute the Laplacian of composed functions. For concreteness, let us assume that $\psi = h \circ G$, $G = (g_1, \ldots, g_k)$ and look

at a single input $x$, i.e. $g_i : \mathbb{R} \to \mathbb{R}$, then

$$(h \circ G)' = \sum_i g_i' \cdot (\partial_i h \circ G), \quad (h \circ G)'' = \sum_i g_i'' \cdot (\partial_i h \circ G) + \sum_{i,j} g_i' \cdot g_j' \cdot (\partial_i \partial_j h \circ G) \quad (14)$$

So at the intermediate layer, we would need to effectively operate with a full Hessian, which is computationally demanding, as has already been observed in [5].

### 3.4 Efficient Gradient-Based Regularization

We would like to derive a (more) tractable strategy for regularizing $\psi$, which (i) avoids the detrimental variance that comes from sampling $\boldsymbol{\xi}$, (ii) does not rely on explicitly convolving the distributions $\mathbb{P}$ and $\mathbb{Q}$, and (iii) avoids the computation of Laplacians as in Eq. (13). Clearly, this requires to make further simplifications. We suggest to exploit properties of the maximizer $\psi^*$ of $F$ that can be characterized by [24]

$$(f^{c'} \circ \psi^*) \, d\mathbb{Q} = d\mathbb{P} \implies \mathbf{E}_{\mathbb{P}}[h] = \mathbf{E}_{\mathbb{Q}}[(f^{c'} \circ \psi^*) \cdot h] \quad (\forall h, \text{ integrable}). \quad (15)$$

The relevance of this becomes clear, if we apply the chain rule to $\triangle(f^c \circ \psi)$, assuming that $f^c$ is twice differentiable

$$\triangle(f^c \circ \psi) = (f^{c''} \circ \psi) \cdot \|\nabla \psi\|^2 + (f^{c'} \circ \psi) \triangle \psi, \quad (16)$$

as now we get a convenient cancellation of the Laplacians at $\psi = \psi^* + \mathcal{O}(\gamma)$

$$F_\gamma(\mathbb{P}, \mathbb{Q}; \psi^*) = F(\mathbb{P}, \mathbb{Q}; \psi^*) - \frac{\gamma}{2} \mathbf{E}_{\mathbb{Q}}\left[(f^{c''} \circ \psi^*) \cdot \|\nabla \psi^*\|^2\right] + \mathcal{O}(\gamma^2). \quad (17)$$

We can (heuristically) turn this into a regularizer by taking the leading terms,

$$F_\gamma(\mathbb{P}, \mathbb{Q}; \psi) \approx F(\mathbb{P}, \mathbb{Q}; \psi) - \frac{\gamma}{2} \Omega_f(\mathbb{Q}; \psi), \quad \Omega_f(\mathbb{Q}; \psi) := \mathbf{E}_{\mathbb{Q}}\left[(f^{c''} \circ \psi) \cdot \|\nabla \psi\|^2\right]. \quad (18)$$

Note that we do not assume that the Laplacian terms cancel far away from the optimum, i.e. we do not assume Eq. (15) to hold for $\psi$ far away from $\psi^*$. Instead, the underlying assumption we make is that optimizing the gradient-norm regularized objective $F_\gamma(\mathbb{P}, \mathbb{Q}; \psi)$ makes $\psi$ converge to $\psi^* + \mathcal{O}(\gamma)$, for which we know that the Laplacian terms cancel [5, 2].

The convexity of $f^c$ implies that the weighting function of the squared gradient norm is non-negative, i.e. $f^{c''} \geq 0$, which in turn implies that the regularizer $-\frac{\gamma}{2} \Omega_f(\mathbb{Q}; \psi)$ is upper bounded (by zero). Maximization of $F_\gamma(\mathbb{P}, \mathbb{Q}; \psi)$ with respect to $\psi$ is therefore well-defined. Further considerations regarding the well-definedness of the regularizer can be found in sec. 7.2 in the Appendix.

## 4 Regularizing GANs

We have shown that training with noise is equivalent to regularizing the discriminator. Inspired by the above analysis, we propose the following class of $f$-GAN regularizers:

---
**Regularized $f$-GAN**

$$F_\gamma(\mathbb{P}, \mathbb{Q}; \psi) = \mathbf{E}_{\mathbb{P}}[\psi] - \mathbf{E}_{\mathbb{Q}}[f^c \circ \psi] - \frac{\gamma}{2} \Omega_f(\mathbb{Q}; \psi)$$

$$\Omega_f(\mathbb{Q}; \psi) := \mathbf{E}_{\mathbb{Q}}\left[(f^{c''} \circ \psi) \|\nabla \psi\|^2\right]$$

(19)

---

The regularizer corresponding to the commonly used parametrization of the Jensen-Shannon GAN can be derived analogously as shown in the Appendix. We obtain,

---
**Regularized Jensen-Shannon GAN**

$$F_\gamma(\mathbb{P}, \mathbb{Q}; \varphi) = \mathbf{E}_{\mathbb{P}}[\ln(\varphi)] + \mathbf{E}_{\mathbb{Q}}[\ln(1 - \varphi)] - \frac{\gamma}{2} \Omega_{JS}(\mathbb{P}, \mathbb{Q}; \varphi)$$

$$\Omega_{JS}(\mathbb{P}, \mathbb{Q}; \varphi) := \mathbf{E}_{\mathbb{P}}\left[(1 - \varphi(\mathbf{x}))^2 \|\nabla \phi(\mathbf{x})\|^2\right] + \mathbf{E}_{\mathbb{Q}}\left[\varphi(\mathbf{x})^2 \|\nabla \phi(\mathbf{x})\|^2\right]$$

(20)

---

where $\phi = \sigma^{-1}(\varphi)$ denotes the logit of the discriminator $\varphi$. We prefer to compute the gradient of $\phi$ as it is easier to implement and more robust than computing gradients after applying the sigmoid.

**Algorithm 1** **Regularized JS-GAN.** Default values: $\gamma_0 = 2.0$, $\alpha = 0.01$ (with annealing), $\gamma = 0.1$ (without annealing), $n_\varphi = 1$

---

**Require:** Initial noise variance $\gamma_0$, annealing decay rate $\alpha$, number of discriminator update steps $n_\varphi$ per generator iteration, minibatch size $m$, number of training iterations $T$
**Require:** Initial discriminator parameters $\omega_0$, initial generator parameters $\theta_0$
  **for** $t = 1, ..., T$ **do**
    $\gamma \leftarrow \gamma_0 \cdot \alpha^{t/T}$  # annealing
    **for** $1, ..., n_\varphi$ **do**
      Sample minibatch of real data $\{\mathbf{x}^{(1)}, ..., \mathbf{x}^{(m)}\} \sim \mathbb{P}$.
      Sample minibatch of latent variables from prior $\{\mathbf{z}^{(1)}, ..., \mathbf{z}^{(m)}\} \sim p(\mathbf{z})$.

$$F(\omega, \theta) = \frac{1}{m} \sum_{i=1}^{m} \left[ \ln\left(\varphi_\omega(\mathbf{x}^{(i)})\right) + \ln\left(1 - \varphi_\omega(G_\theta(\mathbf{z}^{(i)}))\right) \right]$$

$$\Omega(\omega, \theta) = \frac{1}{m} \sum_{i=1}^{m} \left[ \left(1 - \varphi_\omega(\mathbf{x}^{(i)})\right)^2 ||\nabla\phi_\omega(\mathbf{x}^{(i)})||^2 + \varphi_\omega\left(G_\theta(\mathbf{z}^{(i)})\right)^2 ||\nabla_{\tilde{\mathbf{x}}}\phi_\omega(\tilde{\mathbf{x}})|_{\tilde{\mathbf{x}}=G_\theta(\mathbf{z}^{(i)})}||^2 \right]$$

      $\omega \leftarrow \omega + \nabla_\omega\left(F(\omega, \theta) - \frac{\gamma}{2}\Omega(\omega, \theta)\right)$  # gradient ascent
    **end for**
    Sample minibatch of latent variables from prior $\{\mathbf{z}^{(1)}, ..., \mathbf{z}^{(m)}\} \sim p(\mathbf{z})$.

$$F(\omega, \theta) = \frac{1}{m} \sum_{i=1}^{m} \ln\left(1 - \varphi_\omega(G_\theta(\mathbf{z}^{(i)}))\right) \quad \text{or} \quad F_{\text{alt}}(\omega, \theta) = -\frac{1}{m} \sum_{i=1}^{m} \ln\left(\varphi_\omega(G_\theta(\mathbf{z}^{(i)}))\right)$$

    $\theta \leftarrow \theta - \nabla_\theta F(\omega, \theta)$  # gradient descent
  **end for**
  The gradient-based updates can be performed with any gradient-based learning rule. We used Adam in our experiments.

---

## 4.1 Training Algorithm

Regularizing the discriminator provides an efficient way to convolve the distributions and is thereby sufficient to address the dimensional misspecification challenges outlined in the introduction. This leaves open the possibility to use the regularizer also in the objective of the generator. On the one hand, optimizing the generator through the regularized objective may provide useful gradient signal and therefore accelerate training. On the other hand, it destabilizes training close to convergence (if not dealt with properly), since the generator is incentiviced to put probability mass where the discriminator has large gradients. In the case of JS-GANs, we recommend to pair up the regularized objective of the discriminator with the "alternative" or "non-saturating" objective for the generator, proposed in [8], which is known to provide strong gradients out of the box (see Algorithm 1).

## 4.2 Annealing

The regularizer variance $\gamma$ lends itself nicely to annealing. Our experimental results indicate that a reasonable annealing scheme consists in regularizing with a large initial $\gamma$ early in training and then (exponentially) decaying $\gamma$ to a small non-zero value. We leave to future work the question of how to determine an optimal annealing schedule.

# 5 Experiments

## 5.1 2D submanifold mixture of Gaussians in 3D space

To demonstrate the stabilizing effect of the regularizer, we train a simple GAN architecture [20] on a 2D submanifold mixture of seven Gaussians arranged in a circle and embedded in 3D space (further details and an illustration of the mixture distribution are provided in the Appendix). We emphasize that this mixture is degenerate with respect to the base measure defined in ambient space as it does not have fully dimensional support, thus precisely representing one of the failure scenarios commonly

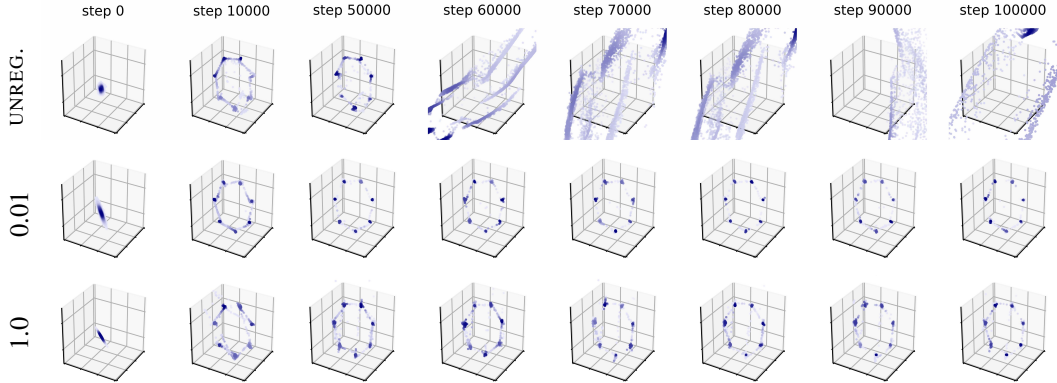

Figure 1: *2D submanifold mixture. The first row shows one of several unstable unregularized GANs trained to learn the dimensionally misspecified mixture distribution. The remaining rows show regularized GANs (with regularized objective for the discriminator and unregularized objective for the generator) for different levels of regularization $\gamma$. Even for small but non-zero noise variance, the regularized GAN can essentially be trained indefinitely without collapse. The color of the samples is proportional to the density estimated from a Gaussian KDE fit. The target distribution is shown in Fig. 5. GANs were trained with one discriminator update per generator update step (indicated).*

described in the literature [3]. The results are shown in Fig. 1 for both standard unregularized GAN training as well as our regularized variant.

While the unregularized GAN collapses in literally every run after around 50k iterations, due to the fact that the discriminator concentrates on ever smaller differences between generated and true data (the stakes are getting higher as training progresses), the regularized variant can be trained essentially indefinitely (well beyond 200k iterations) without collapse for various degrees of noise variance, with and without annealing. The stabilizing effect of the regularizer is even more pronounced when the GANs are trained with five discriminator updates per generator update step, as shown in Fig. 6.

## 5.2 Stability across various architectures

To demonstrate the stability of the regularized training procedure and to showcase the excellent quality of the samples generated from it, we trained various network architectures on the CelebA [17], CIFAR-10 [15] and LSUN bedrooms [32] datasets. In addition to the deep convolutional GAN (DCGAN) of [26], we trained several common architectures that are known to be hard to train [4, 26, 19], therefore allowing us to establish a comparison to the concurrently proposed gradient-penalty regularizer for Wasserstein GANs [10]. Among these architectures are a DCGAN without any normalization in either the discriminator or the generator, a DCGAN with tanh activations and a deep residual network (ResNet) GAN [11]. We used the open-source implementation of [10] for our experiments on CelebA and LSUN, with one notable exception: we use batch normalization also for the discriminator (as our regularizer does not depend on the optimal transport plan or more precisely the gradient penalty being imposed along it).

All networks were trained using the Adam optimizer [13] with learning rate $2 \times 10^{-4}$ and hyper-parameters recommended by [26]. We trained all datasets using batches of size 64, for a total of 200K generator iterations in the case of LSUN and 100k iterations on CelebA. The results of these experiments are shown in Figs. 3 & 2. Further implementation details can be found in the Appendix.

## 5.3 Training time

We empirically found regularization to increase the overall training time by a marginal factor of roughly $1.4$ (due to the additional backpropagation through the computational graph of the discriminator gradients). More importantly, however, (regularized) $f$-GANs are known to converge (or at least generate good looking samples) faster than their WGAN relatives [10].

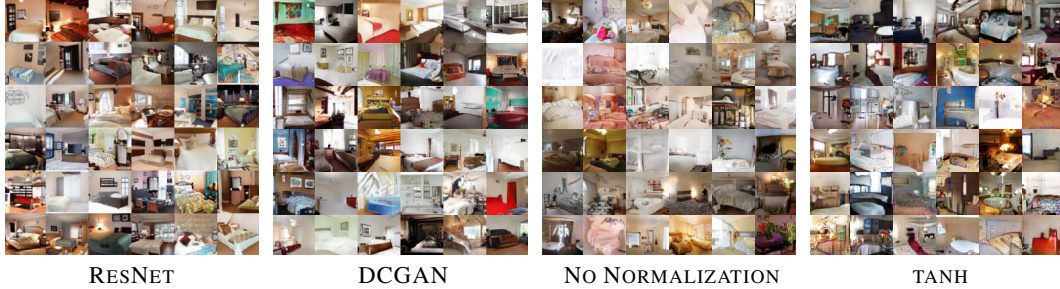

RESNET       DCGAN       NO NORMALIZATION       TANH

Figure 2: *Stability accross various architectures: ResNet, DCGAN, DCGAN without normalization and DCGAN with tanh activations (details in the Appendix). All samples were generated from regularized GANs with exponentially annealed $\gamma_0 = 2.0$ (and alternative generator loss) as described in Algorithm 1. Samples were produced after 200k generator iterations on the LSUN dataset (see also Fig. 8 for a full-resolution image of the ResNet GAN). Samples for the unregularized architectures can be found in the Appendix.*

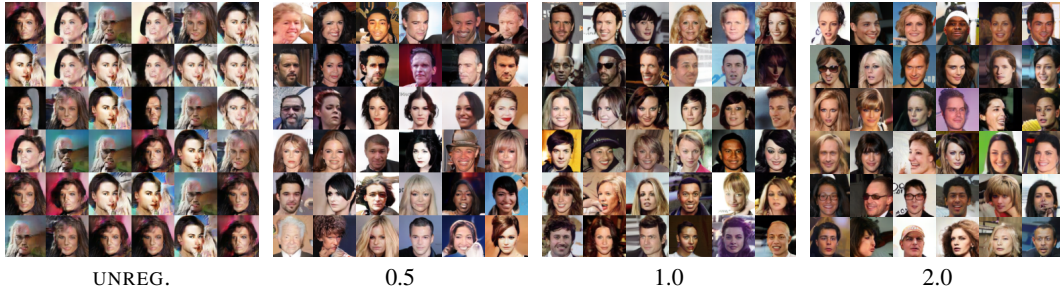

UNREG.       0.5       1.0       2.0

Figure 3: *Annealed Regularization. CelebA samples generated by (un)regularized ResNet GANs. The initial level of regularization $\gamma_0$ is shown below each batch of images. $\gamma_0$ was exponentially annealed as described in Algorithm 1. The regularized GANs can be trained essentially indefinitely without collapse, the superior quality is again evident. Samples were produced after 100k generator iterations.*

## 5.4 Regularization vs. explicitly adding noise

We compare our regularizer against the common practitioner's approach to explicitly adding noise to images during training. In order to compare both approaches (analytic regularizer vs. explicit noise), we fix a common batch size (64 in our case) and subsequently train with different noise-to-signal ratios (NSR): we take (batch-size/NSR) samples (both from the dataset and generated ones) to each of which a number of NSR noise vectors is added and feed them to the discriminator (so that overall both models are trained on the same batch size). We experimented with NSR 1, 2, 4, 8 and show the best performing ratio (further ratios in the Appendix). Explicitly adding noise in high-dimensional ambient spaces introduces additional sampling variance which is not present in the regularized variant. The results, shown in Fig. 4, confirm that the regularizer stabilizes across a broad range of noise levels and manages to produce images of considerably higher quality than the unregularized variants.

## 5.5 Cross-testing protocol

We propose the following pairwise cross-testing protocol to assess the relative quality of two GAN models: unregularized GAN (*Model 1*) vs. regularized GAN (*Model 2*). We first report the confusion matrix (classification of 10k samples from the test set against 10k generated samples) for each model separately. We then classify 10k samples generated by *Model 1* with the discriminator of *Model 2* and vice versa. For both models, we report the fraction of false positives (FP) (Type I error) and false negatives (FN) (Type II error). The discriminator with the lower FP (and/or lower FN) rate defines the better model, in the sense that it is able to more accurately classify out-of-data samples, which indicates better generalization properties. We obtained the following results on CIFAR-10:

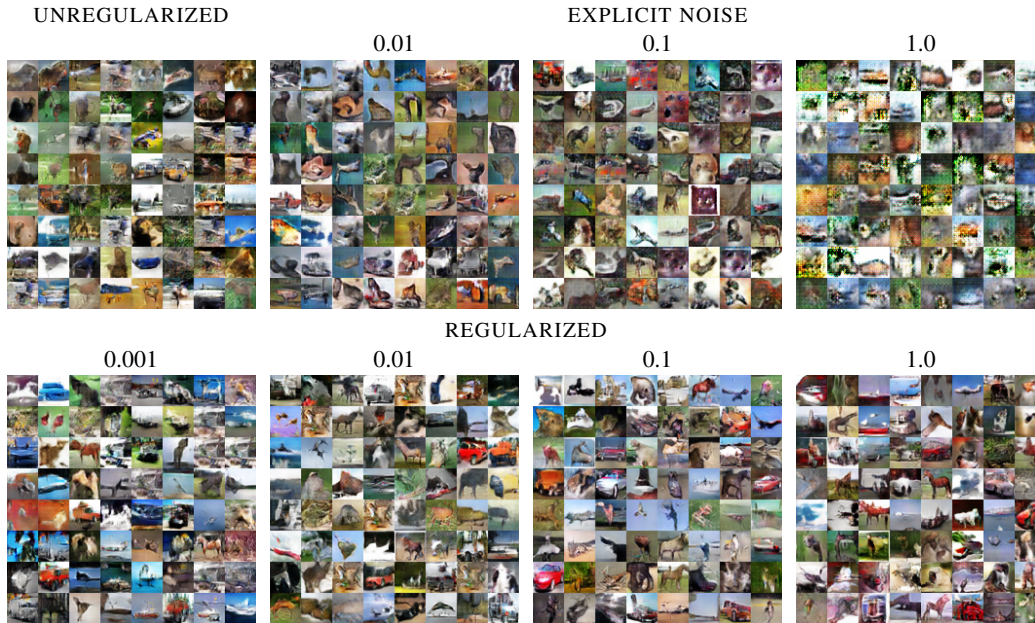

UNREGULARIZED        EXPLICIT NOISE
                0.01            0.1            1.0

REGULARIZED
0.001          0.01           0.1            1.0

Figure 4: *CIFAR-10 samples generated by (un)regularized DCGANs (with alternative generator loss), as well as by training a DCGAN with explicitly added noise (noise-to-signal ratio 4). The level of regularization or noise $\gamma$ is shown above each batch of images. The regularizer stabilizes across a broad range of noise levels and manages to produce images of higher quality than the unregularized variants. Samples were produced after 50 training epochs.*

**Regularized GAN** ($\gamma = 0.1$)

|  |  | True condition | |
| --- | --- | --- | --- |
|  |  | Positive | Negative |
| Predicted | Positive | 0.9688 | 0.0002 |
|  | Negative | 0.0312 | 0.9998 |

Cross-testing: FP: 0.0

**Unregularized GAN**

|  |  | True condition | |
| --- | --- | --- | --- |
|  |  | Positive | Negative |
| Predicted | Positive | 1.0 | 0.0013 |
|  | Negative | 0.0 | 0.9987 |

Cross-testing: FP: 1.0

For both models, the discriminator is able to recognize his own generator's samples (low FP in the confusion matrix). The regularized GAN also manages to perfectly classify the unregularized GAN's samples as fake (cross-testing FP 0.0) whereas the unregularized GAN classifies the samples of the regularized GAN as real (cross-testing FP 1.0). In other words, the regularized model is able to fool the unregularized one, whereas the regularized variant cannot be fooled.

## 6    Conclusion

We introduced a regularization scheme to train deep generative models based on generative adversarial networks (GANs). While dimensional misspecifications or non-overlapping support between the data and model distributions can cause severe failure modes for GANs, we showed that this can be addressed by adding a penalty on the weighted gradient-norm of the discriminator. Our main result is a simple yet effective modification of the standard training algorithm for GANs, turning them into reliable building blocks for deep learning that can essentially be trained indefinitely without collapse. Our experiments demonstrate that our regularizer improves stability, prevents GANs from overfitting and therefore leads to better generalization properties (cf cross-testing protocol). Further research on the optimization of GANs as well as their convergence and generalization can readily be built upon our theoretical results.

## Acknowledgements

We would like to thank Devon Hjelm for pointing out that the regularizer works well with ResNets. KR is thankful to Yannic Kilcher, Lars Mescheder and the dalab team for insightful discussions. Big thanks also to Ishaan Gulrajani and Taehoon Kim for their open-source GAN implementations. This work was supported by Microsoft Research through its PhD Scholarship Programme.

## Footnotes

[1]Code available at https://github.com/rothk/Stabilizing_GANs

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
