[Supplementary Material]

# 7 APPENDIX

## 7.1 Derivation of the Jensen-Shannon Regularizer.

The Jensen-Shannon GAN is typically encountered in one of two equivalent parametrizations: the commonly used "original" GAN parametrization [8],

$$\mathbf{E}_{\mathbb{P}}\left[\ln(\varphi)\right] + \mathbf{E}_{\mathbb{Q}}\left[\ln(1-\varphi)\right] \tag{21}$$

and the Fenchel-dual $f$-GAN parametrization [25], where $f^c = -\ln(1-\exp)$,

$$\mathbf{E}_{\mathbb{P}}\left[\psi\right] - \mathbf{E}_{\mathbb{Q}}\left[f^c \circ \psi\right] = \mathbf{E}_{\mathbb{P}}\left[\psi\right] + \mathbf{E}_{\mathbb{Q}}\left[\ln(1-\exp(\psi))\right] \tag{22}$$

Depending on whether we train the JS-GAN through its Fenchel-dual parametrization or with the original GAN objective we have to either use the regularizer in the general $f$-GAN form in Eq. (19), or in the specific Jensen-Shannon parametrization given in Eq. (20).

We now show how to derive the specific Jensen-Shannon regularizer, which is basically a repetition of the derivation of the general $f$-GAN regularizer presented in the main text. Using the same terminology and following the same line of thought as in section 3.3, i.e. assuming the noise variance is small, we can Taylor approximate the statistics $\varphi$ around $\boldsymbol{\xi} = 0$,

$$\varphi(\mathbf{x}+\boldsymbol{\xi}) = \varphi(\mathbf{x}) + [\nabla\,\varphi(\mathbf{x})]\,\boldsymbol{\xi} + \frac{1}{2}\,\boldsymbol{\xi}^T[\nabla^2\,\varphi(\mathbf{x})]\,\boldsymbol{\xi} + \mathcal{O}(\boldsymbol{\xi}^3) \tag{23}$$

Expanding the noise convolved version of the objective in Eq. (21) and making use of the zero-mean and uncorrelatedness properties of the noise distribution, as well as applying the chain rule, we obtain

$$F_\gamma(\mathbb{P}, \mathbb{Q}; \varphi) = \mathbf{E}_{\mathbb{P}}\left[\ln(\varphi)\right] + \mathbf{E}_{\mathbb{Q}}\left[\ln(1-\varphi)\right]$$
$$+ \frac{\gamma}{2}\left\{\mathbf{E}_{\mathbb{P}}\left[\frac{1}{\varphi}\triangle\varphi - ||\nabla\ln(\varphi)||^2\right] + \mathbf{E}_{\mathbb{Q}}\left[-\frac{1}{1-\varphi}\triangle\varphi - ||\nabla\ln(1-\varphi)||^2\right]\right\} + \mathcal{O}(\gamma^2). \tag{24}$$

Following the same arguments as in section 3.4, one can again show that the Laplacian terms cancel at $\varphi = \varphi^* + \mathcal{O}(\gamma)$ and we arrive at

$$F_\gamma(\mathbb{P}, \mathbb{Q}; \varphi) = \mathbf{E}_{\mathbb{P}}\left[\ln(\varphi)\right] + \mathbf{E}_{\mathbb{Q}}\left[\ln(1-\varphi)\right]$$
$$- \frac{\gamma}{2}\left\{\mathbf{E}_{\mathbb{P}}\left[||\nabla\ln(\varphi)||^2\right] + \mathbf{E}_{\mathbb{Q}}\left[||\nabla\ln(1-\varphi)||^2\right]\right\} + \mathcal{O}(\gamma^2), \tag{25}$$

allowing us to read off the corresponding JS regularizer,

$$\Omega_{JS}(\mathbb{P}, \mathbb{Q}; \varphi) = \mathbf{E}_{\mathbb{P}}\left[||\nabla\ln\varphi(\mathbf{x})||^2\right] + \mathbf{E}_{\mathbb{Q}}\left[||\nabla\ln(1-\varphi(\mathbf{x}))||^2\right]$$
$$= \mathbf{E}_{\mathbb{P}}\left[\frac{1}{(\varphi(\mathbf{x}))^2}||\nabla\varphi(\mathbf{x})||^2\right] + \mathbf{E}_{\mathbb{Q}}\left[\frac{1}{(1-\varphi(\mathbf{x}))^2}||\nabla\varphi(\mathbf{x}))||^2\right] \tag{26}$$
$$= \mathbf{E}_{\mathbb{P}}\left[\frac{1}{(\varphi(\mathbf{x}))^2}||\sigma'(\phi(\mathbf{x}))\nabla\phi(\mathbf{x})||^2\right] + \mathbf{E}_{\mathbb{Q}}\left[\frac{1}{(1-\varphi(\mathbf{x}))^2}||\sigma'(\phi(\mathbf{x}))\nabla\phi(\mathbf{x})||^2\right]$$

In order to obtain the regularizer in the logit-parameterization, with $\varphi(\mathbf{x}) = \sigma(\phi(\mathbf{x}))$, we make use of the following property of the sigmoid

$$\sigma'(t) = \sigma(t)(1-\sigma(t)) \tag{27}$$

which allows us to write

$$\Omega_{JS}(\mathbb{P}, \mathbb{Q}; \phi) = \mathbf{E}_{\mathbb{P}}\left[(1-\varphi(\mathbf{x}))^2||\nabla\phi(\mathbf{x})||^2\right] + \mathbf{E}_{\mathbb{Q}}\left[\varphi(\mathbf{x})^2||\nabla\phi(\mathbf{x})||^2\right] \tag{28}$$

Let us finally also show that these two parametrizations are indeed equivalent at the optimum:

$$f^{c''}(t) = \frac{\exp(t)}{(1-\exp(t))^2} \implies f^{c''}(\ln(\varphi^*)) = \frac{\varphi^*}{(1-\varphi^*)^2} = \frac{\varphi^*}{1-\varphi^*} + \frac{\varphi^{*2}}{(1-\varphi^*)^2} \tag{29}$$

Thus,

$$f^{c''}(\ln(\varphi^*))||\nabla\ln(\varphi^*)||^2 d\mathbb{Q} = \left(\frac{\varphi^*}{1-\varphi^*} + \frac{\varphi^{*2}}{(1-\varphi^*)^2}\right)||\nabla\ln(\varphi^*)||^2 d\mathbb{Q} \tag{30}$$
$$= ||\nabla\ln(\varphi^*)||^2 d\mathbb{P} + ||\frac{\varphi^*}{1-\varphi^*}\nabla\ln(\varphi^*)||^2 d\mathbb{Q} \tag{31}$$
$$= ||\nabla\ln(\varphi^*)||^2 d\mathbb{P} + ||\nabla\ln(1-\varphi^*)||^2 d\mathbb{Q} \tag{32}$$

## 7.2 Further Considerations Regarding the Regularizer

Following-up on our discussion of the regularizer in section 3.4: Due to a support or dimensionality mismatch we may have $\sup F(\mathbb{P}, \mathbb{Q}; \psi) = +\infty$ and $\psi^*$ may not exist. However, with $\psi_\epsilon^*$ being the maximizer of $F_\epsilon$ (which is guaranteed to exist for any $\epsilon > 0$), we get

$$F_\gamma(\mathbb{P} * \mathcal{N}(\mathbf{0}, \epsilon\mathbf{I}), \mathbb{Q} * \mathcal{N}(\mathbf{0}, \epsilon\mathbf{I})) = F_\epsilon(\mathbb{P}, \mathbb{Q}; \psi_\epsilon^*) - \frac{\gamma}{2}\Omega_f(\mathbb{Q} * \mathcal{N}(\mathbf{0}, \epsilon\mathbf{I}); \psi_\epsilon^*) + \mathcal{O}(\gamma^2). \quad (33)$$

As $\epsilon \to 0$, $F_\epsilon$ may diverge and so may $\|\nabla\psi_\epsilon^*\|$. The sequence of regularizers $\Omega_f(\mathbb{Q} * \mathcal{N}(\mathbf{0}, \epsilon\mathbf{I}); \cdot)$, however, converges (at least pointwise) to a well defined limit, which is $\Omega_f(\mathbb{Q}; \cdot)$. This shows that the regularizer is well-defined even under dimensional misspecifications.

We can also justify the approximation in Eq. (18) more rigorously. Starting from the Taylor approximation of $f^{c'}$ at $\psi^*$, we get pointwise

$$f^{c'} \circ \psi = f^{c'} \circ \psi^* + f^{c''} \circ (\psi - \psi^*) + \mathcal{O}(|\psi - \psi^*|^2) \quad (34)$$

So in first order of $\|\psi - \psi^*\|_\infty$, the approximation error is $\Delta := \mathbf{E}_\mathbb{Q}\left[(f^{c''} \circ (\psi - \psi^*)) \cdot \triangle\psi\right]$. Using Green's identity, we can derive the following bound, $\Delta \leq \mathcal{O}(L\delta)$, where we assume $|f^{c'''}| \leq L$ and $\|\nabla\psi_* - \nabla\psi\|_\infty < \delta$. However, as this result only gives a formal guarantee for $(\psi, \nabla\psi)$ sufficiently close to $(\psi^*, \nabla\psi^*)$, we refrain from presenting further technical details.

## 7.3 2D Submanifold Mixture of Gaussians in 3D Space

**Experimental Setup.** The experimental setup is inspired by the two-dimensional mixture of Gaussians in [20]. The dataset is constructed as follows. We sample from a mixture of seven Gaussians with standard deviation $0.01$ and means equally spaced around the unit circle. This 2D mixture is then embedded in 3D space $(x, y) \to (x, y, 0)$, rotated by $\pi/4$ around the axis $(1, -1, 0)/\sqrt{2}$ and translated by $(1, 1, 1)/\sqrt{3}$. As illustrated in Fig. 5, this yields a mixture distribution that lives in a tilted 2D submanifold embedded in 3D space. It is important to emphasize that the mixture distribution is by design degenerate with respect to the base measure in 3D as it does not have full dimensional support. This precisely represents the dimensional misspecification scenario for GANs that we aim to address with our regularizer.

Figure 5: 2D submanifold mixture of Gaussians in 3D, obtained by embedding a two-dimensional mixture with means equally spaced around the unit circle in 3D ambient space. For visualization purposes, the standard deviation of the shown mixture components is 10x larger than the one used in the experiments. Samples are colored proportional to their density which we estimated from a Gaussian KDE with bandwidth selected using Scott's rule [29].

**Architecture.** The architecture corresponds to the one used in [20] with one notable exception. We use 2 dimensional latent vectors $z$, sampled from a multivariate normal prior, (whereas [20] uses 256 dimensional $z$), as we found lower dimensional latent variables greatly improve the performance of the unregularized GAN against which we compare. We did all experiments also for latent vectors of dimension 15: the obtained results are in accordance with those presented in the main text and below.

Both networks are optimized using Adam with a learning rate of $1e-3$ and standard hyper-parameters. We trained on batches of size 512. Ten batches were generated to produce one image of the mixture at given time steps. The generator and discriminator network parameters were updated alternatively.

Figure 6: *2D submanifold mixture. The first row shows one of several unstable unregularized GANs trying to learn the dimensionally misspecified mixture distribution. The remaining rows show regularized GANs (with regularized objective for the discriminator and unregularized objective for the generator) for different levels of regularization $\gamma$. Even for small but non-zero noise variance, the regularized GAN can essentially be trained indefinitely without collapse. The color of the samples is proportional to the density estimated from a Gaussian KDE fit. The target distribution is shown in Fig. 5. GANs were trained with five discriminator updates per generator update step (indicated).*

## 7.4 Image Datasets and Network Architectures

We trained on CelebA [18], CIFAR-10 [15] and LSUN [32]. All datasets were trained on minibatches of size 64. The respective GAN architectures are listed below.

**DCGAN.** For the CIFAR-10 experiments, we used the DCGAN (Deep Convolutional GAN) architecture of [26] implemented in Tensorflow by `https://github.com/carpedm20/DCGAN-tensorflow`.

For the LSUN experiments, we used the DCGAN architecture of [26] implemented in Tensorflow by `https://github.com/igul222/improved_wgan_training`.

In both cases, the discriminator uses batch normalization [12] in all but the first and last layer (except where explicitly stated otherwise). The generator uses batch normalization in all layers except the last one. The latent code is sampled from a 100-dimensional uniform distribution over $[0,1]$ (carpedm20) resp. 128-dimensional normal-(0,1) distribution (Gulrajani).

Both networks were trained using the Adam optimizer [13] (with hyper-parameters recommended by the DCGAN authors) for various different learning rates in the range $[0.001, 0.0001]$. The recommended learning rate $2 \times 10^{-4}$ was found to perform best.

**ResNet GAN.** For the CelebA and LSUN experiments, we used a deep residual network ResNet [11] implemented in Tensorflow by `https://github.com/igul222/improved_wgan_training` (implementation details can be found in [10]).

The ResNets use pre-activation residual blocks with two 3 x 3 convolutional layers each and ReLU nonlinearities. The generator has one linear layer, followed by four residual blocks, one deconvolutional layer and tanh output activations. The discriminator has one convolutional layer, followed by four residual blocks and a linear output layer (the discriminator logits are then fed into the sigmoid GAN loss).

We use batch normalization [12] in the generator and discriminator (except explicitly stated otherwise). Both networks were optimized using Adam with learning rate $2 \times 10^{-4}$ and standard hyperparameters. For further architectural details, please refer to [10] and the excellent open-source implementation referenced above.

## 7.5 Further Experimental Results.

RESNET      DCGAN      NO NORMALIZATION      TANH

Figure 7: *Unregularized GANs (with alternative generator loss) accross various architectures: ResNet, DCGAN, DCGAN without normalization and DCGAN with tanh activations. Samples were produced after 200k generator iterations on the LSUN dataset. Samples for the regularized architectures can be found in the main text. Some of these architectures are known to be hard to train.*

Figure 8: Regularized ResNet GAN. Full-resolution $64 \times 64$ images generated by the regularized ResNet GAN after 200k generator iterations on the LSUN dataset. The initial level of regularization $\gamma_0 = 2.0$ was exponentially annealed to $\gamma_T = 0.01$.

Figure 9: *DCGAN trained with explicitly added noise. Top 3 rows: discriminator and generator trained with noise. Bottom 3 rows: discriminator is trained with noise while generator is trained without noise. White normal noise is added to images from the dataset as well as to samples from the generator during training. The generator was trained through the alternative loss. The level of noise $\gamma$ is shown in the left most column, the different noise-to-signal ratios (NSR) above each column of images. Samples were produced after 50 training epochs.*