[Reviews · NeurIPS 2017]

Reviewer 1



In this paper, the authors have introduced a regularization scheme to train generative models with GANs. They added a penalty on the weighted gradient norm to address the severe failure modes caused by dimensional misspecification between the true and model distributions. In general, this paper is well written and easy to follow. (1) The implementation details of the proposed algorithm are missing. The authors should discuss the implementation details in the main paper. Also, please discuss the trick while training the proposed model. GAN is really hard to train and also difficult to reproduce the results reported by the authors. It will be easier if the authors can include the training tricks in their paper. (2) The training time is not discussed in this paper. There is always a balance between the performance gain and the training time. (3) The experimental part is the weakest part of this paper. The authors may want to include more exciting experimental results in the main paper. I would like to accept this paper if the authors can address the above comments seriously.

Reviewer 2



This paper proposed to stabilize the training of GAN using proposed gradient-norm regularizer. This regularization is designed for conventional GAN, or more general f-GAN proposed last year. The idea is interesting but the justification is a little bit coarse. 1. Eq. (15) is only correct for the maximizer \psi^{\star}. However, the regularizer defined in (21) is for arbitrary \psi, which contradicts this assumption. 2. It seems that increasing the weighting of the regularization term, i.e., \gamma does not affect much the training in Fig. 2. The authors claim that 'the results clearly demonstrate the robustness of the regularizer w.r.t. the various regularization bandwidths'. This is a little bit strange and it will be necessary to show how this regularization can affect the training. 3. For the comparison in the experiments, it is not clear how the explicit noise approach was done. What is the power of the noise? How is the batch-size-to-noise-ratios defined? What's relation between it and the conventional SNR? 4. The results in Section 4.3 make no much sense. From the reported table, the diagonal elements are all quite close to one, which can hardly justifies the advantage of regularized GAN. 5. It will be convincing to compare the proposed method with some other stabilizing training techniques, like Salimans et al 2016 or WGAN technique.

Reviewer 3



In order to solve the problem of requiring careful choice of architecture and parameters in Generative Adversarial Networks (GANs) based deep generative models. The author proposed a new regularization approach which can sovle the problem cause by the dimensional mismatch between the model distribution and the true distribution. By analysis noise covolution, conboblved discriminants and the efficient gradient norm-based regularization the author propose a Gradient-Norm Regularizer for f-GAN. The experiment on MNIST and CIFAR-10 also should that the proposed regularizer is useful. This work is very interesting and enlightening which can help people to build more stable GAN.